# Curvature Adaptable Robotic End-Effectors

*Abstract*— **Flexible robotic manipulators are rapidly gaining traction in automotive assembly to boost productivity and adaptability. Conventional end-effector systems depend heavily on custom tooling engineered for individual curved parts, a strategy that drives up reconfiguration costs, limits interoperability across different product lines, and increases downtime between production runs. We propose an underactuated end-effector that integrates a metasheet of dome-shaped bistable units interconnected into actuation groups, individually addressable via a pneumatic inflation. This arrangement permits transitions between multiple stable configurations, each corresponding to a distinct curvature profile, allowing the manipulator to accommodate different objects found on assembly lines. By tuning the geometry of the proposed end-effector, the system triggers transitions in targeted groups, reconfiguring the system's overall shape to conform to diverse part geometries. This flexibility enables a single manipulator platform to handle a broad family of components without the expense and downtime associated with bespoke tooling changes. By leveraging intrinsic compliance and multistability, the proposed approach strikes an effective balance between mechanical complexity and operational simplicity.**

## I. INTRODUCTION

Automotive production is increasingly challenged by the increasing variety of products (more models, trims, and options) under short cycle times and tight quality constraints. In this setting, curved parts and surfaces make robotic handling and suction gripping especially demanding. Reliable holding depends not only on the suction force, but also on the integrity of the seal, appropriate load paths, and the tolerance to surface variability. A common industrial approach is to deploy part-specific fixtures populated with arrays of conventional suction cups and then commission the gripping layout empirically: an operator adjusts cup engagement points until acceptable sealing and stability are achieved for the specific part and pose. This commissioning step is labor-intensive and fixture-related activities can account for a substantial fraction of investment costs (e.g., 29%) [1]. Shape adaptable suction end effectors capable of reconfiguring on demand to preserve contact quality and force distribution across a variety of parts would provide a major improvement compared to current approaches. However, prevailing morphing solutions often rely on tightly coupled networks of sensors, actuators, and feedback control layered onto compliant substrates. Although these hybrid systems improve shape adaptability, their interdependence can introduce design trade-offs, increase failure modes, and compromise reliability and energy efficiency. Compliance-based morphing structures with inherent reconfiguration capabilities address this issue by embedding the control in the mechanics of the structure itself [2]–[5]. In particular,

engineered arrays of bistable elements provide a discrete set of stable configurations, each corresponding to a specific curvature profile. Transitioning between these states requires only a brief pulse of energy, after which the structure locks passively into its new shape without continuous power draw or intricate closed-loop control. This underactuated paradigm combines mechanical simplicity with functional adaptability, reducing overall component count and simplifying system architecture.

## II. ADAPTABLE SUCTION CUP

We propose an adaptable, underactuated end-effector comprised of a metasheet of dome-shaped bistable units and a flexible lip, actuated through a pneumatic inflation network that generates suction upon contact (Figure 1a). Pneumatic inputs trigger snap-through transitions among multiple programmed stable configurations, each associated with a discrete curvature, enabling the end-effector to conform to objects of varying size and curvature for pick-and-place tasks with minimal control (Figure 1b and Movie 1). At a high level, the concept separates two functions that are typically coupled in conventional suction tooling: (i) grasping, achieved by delivering vacuum to a sealed interface, and (ii) shape switching, achieved by pressurizing the dome array to select a target curvature state. Importantly, once inflation ceases, the bistable elements maintain their new curvature passively, eliminating the need for sensors or continuous energy input. The propose architecture further enables programming different curvature states by reconfiguring the arrangement or geometry of the bistable units, enhancing versatility across industrial applications. A central geometric insight is that inversion thresholds can be programmed through dome geometry, enabling discrete pressure levels to activate subsets of domes selectively. In this implementation, the dome array is partitioned into three groups by assigning three dome heights $H = \{4.0, 4.5, 5.0\}$, which produce representative stable configurations with distinct inversion patterns (filled markers) shown in Figure 1a. By modifying the geometry of the units, we illustrate how a hierarchy of inversion thresholds can be realized using a small number of dome geometries rather than additional actuators, thereby providing a means to realize heavily underactuated shape reconfiguration.

## III. SUCTION GRASPING MODEL

To quickly assess whether a suction cup can achieve a reliable seal on a given surface, we approximate the cup as a quasi-static spring system and compute the deformation energy necessary to sustain contact and maintain the seal [6].

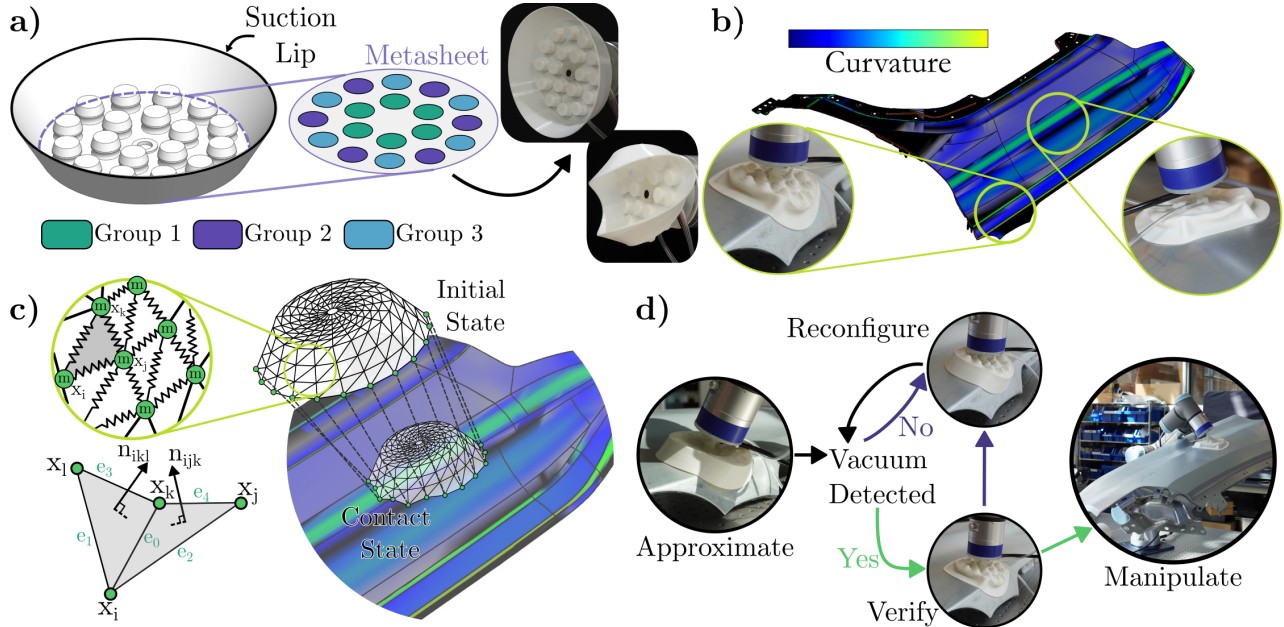

Fig. 1. Curvature-adaptable, underactuated suction end-effector integrating a dome-patterned metasheet with a compliant lip. a) Metasheet of bistable domes units grouped by geometry and interconnected to enable programmable inversion thresholds to achieve a set of discrete curvature states. b) Experimental adaptation to curved automotive panels. c) Spring-lattice model used to evaluate deformation energy and seal feasibility between initial and contact states. d) Discrete manipulation loop that reconfigures curvature states and verifies seal formation via vacuum feedback.

To represent the dome array system, we developed a spring lattice model [7]–[10]. Our model consists of different linear springs arranged to capture the initial geometry of the suction cup (see Figure 1c). Given this, the in-plane strain energy can be written as: $E_s(x_i, x_j) = \frac{1}{2}\left(\frac{\sqrt{3}Et}{2}\right)(\|r_{ij}\| - s_{ij})^2$. Where $r_{ij} = x_i - x_j$ is the vector from node $i$ to node $j$, $s_{ij}$ is the original length of the edge $i - j$, t is the thickness, and E is the elastic modulus. For the discretized bending energy, we utilized a similar expression, which can be written as: $E_B(\theta, \theta_0) = \frac{1}{2}B(\varphi(\theta) - \varphi(\theta_0))^2$. Where $\theta$ is the angle form by the normal vectors of neighboring faces, $B$ is the bending stiffness of the sheet given by $B = \frac{Et^3}{12(1-\nu^2)}$ and $\varphi(\theta)$ is a bending function that can be written as $\varphi(\theta) = 2\tan(\theta/2)$ [11]. The final configuration of the system is obtained by minimizing the total elastic energy, defined as the sum of the contributions of stretching and bending. To consider the stable states, we employ a two-stage optimization procedure. First, we impose boundary conditions on the initially flat region to match the curvature of the stable state and solve for the corresponding equilibrium geometry. This configuration is then used as the initial condition for a second optimization that evaluates the sealing performance. To evaluate the feasibility of the seal, the suction lip nodes are projected onto the surface of the object (see Figure 1c - Contact State). This projection is imposed as a boundary condition on the suction lip, and the final shape is calculated. The seal is guaranteed if none of the springs on the lip are stretched by more than 5% of their original value [6]. This allows us to determine the true capability of the proposed design and compare it with other suction cups.

## IV. TOWARDS FLEXIBLE MANUFACTURING

The end-effector concept is intended to reduce reliance on geometric perception when the target curvature is unknown, for example, when a new part is introduced, when curvature varies across the intended contact region, or when the contact patch cannot be predicted in advance. In this setting, the controller does not attempt to estimate curvature. Instead, it exploits the fact that the end-effector offers a finite set of stable curvature states and evaluates them directly using a simple seal indicator. Considering this, we created a simple manipulation loop (see Figure 1d), where the vacuum source provides suction for gripping, the end-effector is switched among discrete stable states, and a pressure sensor provides the minimal feedback needed to verify whether a seal is achieved and maintained (see Movie 1). This manipulation loop is treated as a discrete decision problem, whereby the metasheet's stable states are switched to increase, the likelihood of forming a perimeter seal under a fixed preload. Consequently, the controller cycles through the available shapes (i.e., states) stopping at the first state that produces a reliable seal. This differs from continuous shape control because the algorithm does not regulate curvature, instead only determining which pre-defined stable configuration is compatible with the local surface, a significantly simpler control problem that preserve the benefits afforded by shape morphing end effectors.

## V. SUPPLEMETARY VIDEO

https://drive.google.com/file/d/
1g8ZTcpyCOaaXjTawNhQkOwJTEZ_K4-zR/view?
usp=sharing

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
