# OpenReview forum: "Curvature Adaptable Robotic End-Effectors"
_IEEE.org/ICRA/2026/Workshop/Manipulation_Robustness — ICRA 2026_

### Official Review · Reviewer_DdiR · 2026-05-06
**Good work on suction cup-based vacuum manipulator**

**Rating:** 8
**Confidence:** 3

**Review:**

Fascinating work on hardware and mechanical intelligence — embedding shape adaptability directly into the mechanical structure via bistable metasheets is a clever way to cut down on the sensing and control overhead typical of industrial grasping systems. The decision to decouple shape reconfiguration from vacuum grasping is a smart call, keeping both functions lean and the overall system surprisingly easy to operate. Programming snap-through thresholds purely through dome geometry is the standout idea here, getting a useful multistable shape space out of just a few dome heights in a way that feels both elegant and practical.

The discrete manipulation loop in Section IV is where this really pays off — skipping curvature estimation and just cycling through stable states until a seal forms is a refreshingly simple strategy that should hold up well in real factory settings. The proof-of-concept experiments on actual automotive panels are convincing, and the supplementary video adds a lot to the overall credibility. Going forward, a quantitative analysis of the workable curvature range and some initial data on fatigue life under repeated snap-through cycling would strengthen the case for broader industrial deployment. It would also be great to see how the system performs on uneven or non-uniform surfaces, as these are common in real assembly environments and would further demonstrate the generality of the approach.

---

### Decision · Program_Chairs · 2026-05-21

Accept